# Obesity and Eating Disorders in Children and Adolescents: The Bidirectional Link

**DOI:** 10.3390/nu13124321

**Published:** 2021-11-29

**Authors:** Stella Stabouli, Serap Erdine, Lagle Suurorg, Augustina Jankauskienė, Empar Lurbe

**Affiliations:** 1First Department of Pediatrics, Hipnmpokration Hospital, Aristotle University, 54124 Thessaloniki, Greece; sstaboul@auth.gr; 2Hypertension and Arteriosclerosis Research and Implementation Center, School of Medicine, Marmara University, Istanbul 34722, Turkey; serap.erdine@gmail.com; 3Tallinn Children’s Hospital, 2813419 Estonia, Estonia; lsuurorg@gmail.com; 4Pediatric Center, Institute of Clinical Medicine, Vilnius University, 01513 Vilnius, Lithuania; augustina.jankauskiene@santa.lt; 5Department of Pediatrics, University of Valencia, 1346010 Valencia, Spain; 6CIBER Fisiopatologia Obesidad y Nutricion, Instituto de Salud Carlos III, 28029 Madrid, Spain

**Keywords:** obesity, eating disorders, children, adolescents

## Abstract

Obesity, eating disorders and unhealthy dieting practices among children and adolescents are alarming health concerns due to their high prevalence and adverse effects on physical and psychosocial health. We present the evidence that eating disorders and obesity can be managed or prevented using the same interventions in the pediatric age. In the presence of obesity in the pediatric age, disordered eating behaviors are highly prevalent, increasing the risk of developing eating disorders. The most frequently observed in subjects with obesity are bulimia nervosa and binge-eating disorders, both of which are characterized by abnormal eating or weight-control behaviors. Various are the mechanisms overlying the interaction including environmental and individual ones, and different are the approaches to reduce the consequences. Evidence-based treatments for obesity and eating disorders in childhood include as first line approaches weight loss with nutritional management and lifestyle modification via behavioral psychotherapy, as well as treatment of psychiatric comorbidities if those are not a consequence of the eating disorder. Drugs and bariatric surgery need to be used in extreme cases. Future research is necessary for early detection of risk factors for prevention, more precise elucidation of the mechanisms that underpin these problems and, finally, in the cases requiring therapeutic intervention, to provide tailored and timely treatment. Collective efforts between the fields are crucial for reducing the factors of health disparity and improving public health.

## 1. Introduction 

Obesity, eating disorders (EDs) and unhealthy dieting practices among children and adolescents are alarming health concerns due to their high prevalence, more than 1 hundred million [1], and adverse effects on physical and psychosocial health. Even when, traditionally, obesity and EDs have been looked at as separate conditions, there is emerging evidence highlighting important overlaps, among others, etiology, comorbidity, risk factors and prevention approaches [2]. Environmental and social factors, weight-related teasing by family or peers, thin beauty ideal perceptions by social environment or media may enable transition from obesity to EDs and vice versa [3]. In the presence of obesity and its cardiometabolic adverse health consequences, possible additional EDs could further trigger the burden of current health status and future outcomes [4]. In the present review, we outline the rationale for the awareness and recognition of risk factors that increase vulnerability of obese children and adolescents to EDs and cover several aspects starting with definitions, common pathogenesis, as well as possible implications on treatment outcomes. Finally, we present the evidence that EDs and obesity can be managed or prevented using the same interventions in the pediatric age.

## 2. Obesity 

Obesity, characterized by the deposition of excessive fat in the body, has been well documented in both sexes, all age groups, and for every geographical and ethnic group. A straightforward method to assess body fat indirectly is body mass index (BMI). According to BMI, weight status is classified in children and adolescents as overweight >85th percentile to <95th percentile and obesity 95th percentile or greater [5]. The WHO recommends the use of BMI z-score defining overweight as having a BMI z-score >1 but less than 2 and obesity z-score as having a BMI z-score equal to or >2 [6]. 

The prevalence of overweight and obesity among children has increased substantially worldwide since the 1990s. According to the WHO, in 2016, one hundred and twenty-four million children and youth between 5 and 19 years of age were obese and 41 million under the age of 5 were overweight or obese [1]. Childhood obesity is more prevalent in developed countries, although an upwards trend is also seen in developing countries [1]. This issue deserves more attention due to the long-term health effects it may bring on, including obesity persisting into adulthood and increased risk of chronic diseases. Immediate and long-term psychosocial health consequences can also be present, including the potential for reduced self-esteem and depression. 

Linked to the development of obesity is the interaction among environmental, behavioral, genetic and metabolic factors [7]. This complex interaction leads to a multifactorial chronic disease with a variety of phenotypes and clinical presentations. All of these combined explain the difficulties in management and treatment responses [8]. Factors contributing to the rise in obesity prevalence world-wide mostly focus on environmental and behavioral elements. Changes in the child’s environment in terms of easy affordability of high-calorie fast food, increased portion size, intake of sugar-sweetened beverages (SSBs) and a sedentary lifestyle are associated with increased incidence of obesity [9]. 

One of the extensively studied causes for obesity is dietary patterns. Even in early life, feeding patterns have been linked to an increased incidence of obesity. In an observational study by Gillman et al. [10], it was reported that in pre-school age children of mothers who did not smoke or gain excessive weight during pregnancy, breast fed for 12 months, and slept 12 h/day, presented an obesity prevalence of 6% at age 3 years compared to a prevalence of 29% among children with the opposite of these four mother/child behaviors. Diet pattern and quality are important issues in the development of obesity. Considering diet, it is important to name discretionary food, a relevant element contributing to childhood obesity. One of the typical examples of this kind of food is SSBs containing a high amount of sugar [11]. In a birth cohort followed from age 2 to 17, investigators reported a significant association between SSBs consumption and increasing BMI z-scores [12]. 

Together with diet, the other pivotal element is physical activity. Advancements in technology have contributed to more sedentary behavior in children and adolescents. Screen time includes time spent viewing television, computer use, playing electronic games, and using mobile phones. Currently, screen time is the most common sedentary behavior, starting even in infancy [13]. Time spent on screen-based activities can replace time for physical activity and may affect physical and mental health in youth [14]. Adverse effects of excessive screen time on physical strength, obesity, and sleep disturbances have been documented in many studies [15]. Sleep disturbance is a commonly overlooked risk factor associated with high BP in children and adolescents. Lower levels of parental education, regular enforcement of rules about caffeine, and presence of electronics in the child’s bedroom overnight are among the factors related to poor sleep [16]. Along with the above well-recognized factors, the presence of socioeconomic adversity, family dysfunction, offspring distress and junk food should be considered [17].

The role of genetics and its contribution to obesity has been filled with a large amount of research. The susceptibility of weight gain varies among individuals, suggesting that there is a heritable component of obesity that interacts with environmental factors [18]. Considering genetic factors, most cases of obesity are polygenic in nature, with multiple genes making small contributions to the overall phenotype. Therefore, genetic susceptibility may affect weight when coupled with other contributing environmental and behavioral factors. In contrast, monogenic obesity is uncommon, accounting for 3 to 5% of obese children, presenting early weight gain often between the first and second year of life [18]. A mutation in the melanocortin 4 receptor gene (MC4 R) is the most common gene defect, which is associated with a severe, early form of obesity in children [19]. 

Adverse childhood experience (ACE) and its link with obesity has received more attention in the last years. In a recent study, children who had high intrafamilial adversity scores were more prone to be obese than children with low scores [20]. These results are in agreement with the findings of a meta-analysis of 41 studies in which the association of child maltreatment and obesity was assessed [21]. Nowadays, ACE is known to be a potentially modifiable risk factor for obesity.

## 3. Eating Disorders

Disordered eating behaviors and EDs both cover a broad group of dimensional maladaptive cognitions and behaviors relating to eating and weight, but differ in their diagnosis [22]. Eating disorders refer to psychiatric disorders characterized by abnormal eating or weight control behaviors [23]. According to the Diagnostic and Statistical Manual of Mental Disorders Fifth Edition, specific EDs include anorexia nervosa, bulimia nervosa (BN) and binge eating (BE) [24]. Although the prevalence of EDs varies according to study populations and the criteria used to define them [25], they are of great concern given their serious health consequences that may lead to significant impairments in health, psychosocial functioning, and quality of life [26]. The onset of EDs is usually during adolescence, with the highest prevalence in girls, but EDs may be present in children as young as 5 to 12 years [27]. Recognition of EDs may help to prevent obesity or help weight loss in cases of sustained obesity [28]. Eating disorders may accompany childhood and adolescent obesity or may evolve after intensive interventions to treat obesity.

Putative risk factors for EDs have been investigated, testing a wide range of environmental [29,30,31] and genetic factors [32,33]. A recent umbrella review of published meta-analyses, including 50 associations from nine meta-analyses, found evidence for childhood sexual abuse as a risk factor for BN and appearance-related teasing victimization for any ED [34]. There were no ED risk factors supported by convincing evidence possibly due to the small number of large-scale collaborative longitudinal studies assessing the relationship between conditions preceding the onset of the disorder and the development of EDs [34]. 

A new element has come into play, which is food insecurity, characterized by limited or uncertain means of accessing nutritious food in a safe and socially acceptable manner. Emerging evidence consistently indicates that food insecurity is cross-sectionally associated with the bulimic-spectrum among adults. This has been shown in a national representative sample of US adults. During a 12-month period, diagnoses of bulimic-spectrum disorders, mood disorders, and anxiety disorders were more common among individuals who have experienced food insecurity than among those who were food secure. The study highlighted that the greatest difference was observed for bulimic-spectrum eating disorders [35]. Considering these findings, it may be necessary to take the pediatric population into consideration. This emerging evidence needs much more research to better understand this issue.

Whether ACEs are true risk factors for the development of eating disorders remains unclear due to the scarcity of not only prospective studies but also the potential selection bias in clinical samples. In currently available studies, inconsistent results have been reported. In a population-based study, the authors stated that experiences of life events are associated with specific eating behaviors in children aged 10 years [36]. These findings sustain the fact that a link between adverse life events and emotional overeating exists [37,38].

## 4. Links between Obesity and EDs 

In the presence of obesity in the pediatric age, disordered eating behaviors are highly prevalent, increasing the risk of developing EDs. The EDs that have been most frequently observed in subjects with obesity are BN and BE, both of which are characterized by abnormal eating or weight-control behaviors [39,40]. The typical characteristics of BN and binge-eating disorder (BED) are recurrent BE episodes, defined as losing control over eating amounts of food that are objectively large. While attempts to prevent weight gain through inappropriate compensatory behaviors such as self-induced vomiting is characteristic of BN, BED does not share this characteristic [40,41]. 

Both obesity and EDs are generally studied and treated as independent disorders; however, obesity and EDs can have a bidirectional impact. Among the different EDs, BE is the one with the highest prevalence of comorbid obesity, followed by BN, and almost 30% of female patients with EDs had lifetime obesity [42].

Several mechanisms linking obesity with EDs and vice versa have been proposed, among other environmental and individual risk factors.

### 4.1. Environmental Risk Factors

Some specific socioenvironmental conditions may act as common risk factors for EDs and obesity. The most common are family and peer teasing, perceived social pressure, frequent criticism or bullying [43]. Also, images on television or social media focus on the ideals of slimness and beauty contribute to body dissatisfaction [44,45], which can act as a risk factor. Beside the most common factors, many others can be related with family BE behaviours, parental mood, anxiety or substance use disorder, family dissonance, high parental demands or perfectionism, and parental separation, identified as possibilities that play a role in the onset of obesity and EDs. Finally, traumatic life events and negative childhood experiences (sexual and physical abuse) also increase the risks [46].

### 4.2. Individual Risk Factors

#### 4.2.1. Biological-Genetic Risk Factors

The strongest known susceptibility locus for obesity is the fat mass and obesity-associated (FTO) gene [47,48,49]. Even though it is not fully clear how FTO variants influence obesity, FTO associations with several EDs, including BED, are apparent [50]. Indeed, variants of the FTO gene are associated with poor behavioural regulation and BED, suggesting a genetic role in the pathogenesis of this disorder [50]. Genetic factors notably influence the regulation of neural circuits by controlling the appetite and satiety pathways, as well as the regulation of brain reward systems. Single-Nucleotide Polymorphisms in genes linked to hypothalamic appetite and satiety mechanisms may be involved in the development of EDs related to obesity such as BED and BN [51]. 

#### 4.2.2. Psychological and Personality Risk Factors

Some specific psychological characteristics such as low self-esteem, negative self-evaluation, and high body dissatisfaction may contribute to the development of EDs and obesity [52]. Research suggests a link between emotional regulation and BE, as well as food addiction [53,54]. When negative affect and emotional dysregulation precede the occurrence of BE episodes, it exacerbates guilt and shame, creating a vicious cycle of losing control over eating [55]. 

#### 4.2.3. Neuropsychological and Brain Activity Risk Factors 

The brain is central to basic research, prevention and treatment in the context of obesity and EDs [56]. Until very recently, little was known about the neuropsychological mechanisms of EDs and obesity. Mesocorticolimbic mechanisms that increase “liking” include brain hedonic hotspots, specific subregions that can causally increase the hedonic effect of palatable tastes. In contrast, a much larger mesocorticolimbic circuit generates the motivation to “want” or induce to obtain and consume food rewards [57].

Theorists focused on the reward circuit because eating palatable food increases activation in reward-related regions, including the ventral and dorsal striatum, midbrain, amygdala, and orbitofrontal cortex, and causes dopamine release in the dorsal striatum in both humans and other animals [58]. Functional, molecular and genetic neuroimaging has highlighted the existence of brain abnormalities and neural fragility factors associated with obesity and EDs, such as overeating or anorexia nervosa [58]. A better understanding of wanting and linking mechanisms tailored to individual types of EDs and obesity could lead to better therapeutic strategies, and perhaps help people who wish to more effectively create stop signals to their own needs [57].

#### 4.2.4. Behavioural Risk Factors 

Body dissatisfaction is a well-documented psychological aspect of obesity, especially for women, and research with female college students found that lifetime experiences of weight stigma significantly mediated the relationship between BMI and body dissatisfaction [55].

Diet is the most significant behavioural risk factor for the onset of BED. It is well documented that dieting increases the risk of overeating to counter calorie deprivation and executive function, weight gain over time [42].

Social isolation can be inherently stressful, depressing, and anxiety-provoking. To heal these distressing feelings, an individual can engage in emotional eating, where the food serves as a source of comfort. This has gained special attention during the COVID pandemic [59,60].

#### 4.2.5. Biochemical

Eating behaviour is a complex process controlled by the neuroendocrine system, of which the hypothalamic–pituitary–adrenal axis (HPA axis) is the main component and dysregulation of the HPA axis has been associated with EDs [61,62]. 

Serotonin also has an inhibitory executive function on eating behaviour [63]. Several studies have assessed the relationship between the noradrenergic system and EDs. A recent systematic review identified a series of key data on the relationship between the noradrenergic system and EDs. Besides its relevant direct, hypothalamus-based actions on feeding regulation, the noradrenergic system is indirectly implied in various endocrine networks controlling human nutrition [64]. Dopamine is a neurotransmitter that regulates the rewarding nature of food [65]. Neuropeptide Y is a hormone that promotes eating and reduces metabolic rate [66]. Leptin has an inhibitory executive function that affects appetite by inducing a feeling of satiety [67]. Ghrelin is an appetizing hormone produced in the stomach and upper part of the small intestine [68]. Circulating leptin and ghrelin levels are an important factor in weight control. Although often associated with obesity, both hormones and related executive functions have been implicated in the pathophysiology of anorexia nervosa and BN [67,68]. 

#### 4.2.6. Gut Bacteria and Immune System

There is increasing interest in the association of gut bacteria with diseases such as diabetes, obesity, inflammatory bowel disease, and psychiatric disorders. The gut microbiota influences nutrient fermentation, body weight regulation, gut permeability, hormones, inflammation, immunology, and behaviour (gut–brain axis) [69].

The gut microbiome plays a vital role, not only in regulating mood and behaviour, but also in regulating metabolic function, appetite control and weight [70]. Studies have shown that most patients with anorexia and BN have elevated levels of autoantibodies that affect hormones and neuropeptides that regulate appetite control and stress response. A link between the gut microbiome and EDs affecting up to 10 percent of the population has been shown [71].

## 5. Management 

Evidence-based treatments for obesity and EDs in childhood include as first-line approaches weight loss with nutritional management and lifestyle modification via behavioral psychotherapy, as well as treatment of psychiatric comorbidities if those are not a consequence of the ED [27]. The majority of children and adolescents under supervised obesity treatment may have improvements or no change to ED risk profiles [72]. Higher baseline dietary restraint scores in obese children have been associated with increased rates of premature drop out from the intervention program compared to children who completed the program, independent of gender, age, and BMI z-score at baseline and mother’s education level [73]. On the other hand, in a secondary analysis of an RCT focusing on changes in energy intake and diet quality during obesity treatment with post-treatment eating pathology in adolescents, there was no association between intensity of diet and EDs [74]. In a systematic review, current measures of dietary restraint and dieting are not associated with ED risk in the short term; however, long-term data are limited [75].

### 5.1. Weight Loss, Diet, Behavioral Therapy, Lifestyle Modification

Most organizations support weight maintenance or weight loss as a treatment goal for the management of pediatric obesity. Lifestyle intervention programs for youth with some degree of overweight recommend considering a wide multidimensional approach covering eating and dietary habits. Suffering from weight-related teasing during childhood and adolescence might lead to emotional eating which, in turn, could impair long-term weight loss maintenance [76]. Even when programs aiming to treat shared risk factors did not result in significant differences in terms of weight status, it had an impact on body dissatisfaction, dieting and weight-control behaviors [22]. 

Cognitive behavioral therapy (CBT) emphasizes on restructuring of the harmful patterns that infiltrate daily functioning and changing habits and attitudes that maintain psychological disorders. CBT has been suggested as a promising treatment approach for EDs and obesity [77]. However, CBT would be considered as a second-line option when family-based multicomponent behavioral weight loss treatment (FBT) has not been effective or could not be applied [77,78].

Multicomponent interventions are regarded to have higher rates of weight loss. FBT is considered effective at treating childhood obesity and a treatment option for disordered eating and obesity in children [79]. Compared to an adolescent-focus intervention, a healthy family-based lifestyle modification could result in increased sustainably of changes [27]. The results of a clinical trial including adolescents on 4-month FBT and subsequent 8-month weight maintenance interventions showed that weight change following FBT and maintenance were reported to be independent of concurrent physiopathology and EDs in the short or long term [80].

### 5.2. Motivational Interviewing 

Frequent counseling may be required in order to help patients maintain motivation to achieve a healthy weight [81]. Motivational interviewing (MI) focus on engagement by establishing a working relationship with the patient in order to explore and plan the need for changes, while at the same time avoiding stigmatizing language regarding weight that may negatively impact a teen and result in BE, decreased physical activity, social isolation, avoidance of health care services, and increased weight gain [27]. Effective health provider–patient communication using MI techniques have been proved useful to encourage positive behavior changes [5]. 

Even when there is a constant interest for a better approach to prevent and treat obesity among the youth, the actual population-oriented interventions and traditional medical care have not had the expected impact. This shows us that new alternatives are needed in order to fight obesity effectively. As an example, a personalized approach and more intense family-based multi-professional weight management called “*Personalized approach in obesity management*” was initiated in Estonia, being a successful long-term project for dealing with overweight children. Using the motivational interview method, the self-motivation of parents and the child for lifestyle changes was examined using the LINE chair Visual Analogue Scale (VAS—1–10 points), real goals for the child’s lifestyle change were selected [82]. According to self-assessment, only 14% of children had similar aspects of health compared to healthy children and, as for the parents, the corresponding figure was 9%. Quality of life estimated that the indicator of children’s physical and emotional health is the most frequently disturbed (in 90–92% of respondents). The relevance of the project’s results can be seen by its nomination as an example of best practice in Estonia in the EU Joint Action on Nutrition and Physical activity (JANPA) [83].

### 5.3. APPs and MHealth

Feasibility of medicine-based mHealth intervention targeted for adolescents have been assessed in order to increase high retention and adherence rates in weight loss interventions [84]. The intervention using new technologies, mHealth, may contribute to reduce the degree of overweight in a more cost-effective manner compared to the classical intervention at the clinic [84].

### 5.4. Public Health Approaches

One of the main risk factors for disordered eating is body dissatisfaction. However, most countries’ public health approaches to confront overweight and obesity frequently use messages that may increase body dissatisfaction in children and adolescents [85]. Thus, it remains a challenge for policy makers to balance between sociocultural pressure for thinness and obesogenic environment. The role of the schools at the time to promote health and educational attainment has been highlighted by the World Health Organization’s (WHO) Health Promoting Schools (HPS) framework. In a Cochrane review, the WHO HPS framework was found to be effective at improving aspects of student health including BMI z-score at the population level [86]. 

### 5.5. Drugs

In combination with lifestyle interventions, the only medication for the treatment of obesity in youth 12 years and older approved by FDA is Orlistat. However, it should be used with caution in the presence of ED psychopathology to avoid misuse as a purging agent [87,88]. Orlistat is the only available anti-obesity drug that does not involve the mechanisms of appetite. It induces weight reduction via the inhibition of lipases in the mucous membranes of the stomach, small intestine, and pancreas, thereby preventing the breakdown of triglycerides into fatty acids and their absorption in the intestines [88]. 

### 5.6. Bariatric Surgery

Few studies have assessed the effect of bariatric surgery on disordered eating symptoms. In a study that included 19 adolescents with severe obesity who underwent a reversible bariatric procedure, improvements of emotional and behavioral factors were documented [89]. In a sub-study of the Teen-LABS Consortium, the application of bariatric surgery in adolescents demonstrated better outcomes 1 year after, not only in weight reduction but also in disordered eating symptoms, as compared to those who were under the lifestyle modification program [90]. Participants in the Adolescent Morbid Obesity Surgery (AMOS) study, 5 years of follow-up after Roux-en-Y gastric bypass surgery showed that BE and uncontrolled eating were moderately improved at the end of the period [91]. A small decrease in emotional eating and a small increase in cognitive restraint were also noted between baseline and 5 years after surgery. Higher scores for BE and emotional eating at Lang 2 years and 5 years, and for uncontrolled eating at 2 years after surgery, were also significantly associated with smaller percentage changes in BMI at 5 years relative to baseline. These data suggest that bariatric surgery alone does not improve adolescents’ eating behavior and the need for a multidisciplinary team for long-term health support after adolescent bariatric surgery [91]. 

### 5.7. Screening for Adverse Childhood Experiences

There are limited data on the effect of structured intervention models to treat obesity and EDs in children who have experienced ACEs. Since there is an association of ACEs with obesity and EDs, the healthcare team should consider the possibility of having an ACE. Screening for ACEs would be regularly performed using validated tools [92,93]. In the case of an affected child, multicomponent intervention strategies should include appropriate psychosocial support and counseling to manage anxiety due to trauma that would impede the effectiveness of obesity and ED treatment. Of note, prevention policies for ACEs that may chance self-confidence, social and emotional skills could accompany healthy eating education on an individual and family level.

## 6. Conclusions

The rising tide of obesity and EDs and the link between them outlines the rationale for awareness and recognition of amenable risk factors that increase vulnerability. The importance of early detection is unquestionable and pediatricians are in a unique position to identify early and disrupt their progression. Despite well-researched links between the physical and mental health of youth in the presence of obesity, the resulting mental health toll is largely ignored. The importance of identifying risk factors shared by both obesity and EDs may serve as an important focal point for an intervention aimed at simultaneously addressing both of them. Obesity and EDs are important health challenges in children and adolescents; therefore, future research is necessary for early detection of risk factors for prevention, more precise elucidation of the mechanisms that underpin these problems and, finally, in the cases requiring therapeutic intervention, to provide tailored and timely treatment. Collective efforts between the fields are crucial for reducing the factors of health disparity and improving public health.

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
