# Peer review of "Obesity and Eating Disorders in Children and Adolescents: The Bidirectional Link"

_nutrients, 2021, doi:10.3390/nu13124321_

Round 1

Reviewer 1 Report

This review has significant promise however, it also needs a significant amount of work. Below are my comments

Introduction 

  1. I understand that the prevalence of obesity and ED is high, but it would be great if you can provide statistics and cite them (i.e. ~165 million kids obese in the world). I know you mention it later in the manuscript, but it would be great to have it here as well)
  2. In the introduction you talk about kids and adolescents. You should outline what ages you are covering in this review. I know you say pediatric but maybe a parenthetical reference would be great.
  3. Cite lines 38-39

Obesity

  1. Cite lines 68-70
  2. Cite lines 8-86
  3. In sociology and psychology literature socioeconomic adversity and family dysfunction have been linked fairly consistently to obesity. I think considering them as less conventional is minimizing the contributions of the two fields
  4. You should also examine the association between ACEs and obesity as you refer to some of them later in the ED section. 
  5. There need to be more citations in this section. You make several claims throughout this section that are not cited.

ED

Overall this section was better cited than the obesity section. I would suggest investigating work that has examined ACEs a little more closely in this section.

Link between obesity and ED

Environmental risk factors

This is a poorly cited section with multiple claims but no cited evidence to back it up.

Biological-genetic risk factors

Please provide examples of the genes that overlap

Psychological and personality risk factors

What are the different personality traits that may play a role? You state a large body of research suggests a link, please provide evidence

Neuropsychoogical and brain activity risk factors

Entire second paragraph is not cited

Behavioral and risk factors

Same as above. Poorly cited. Provide examples of where this happens (studies)

Biochemical

Same as above section. It needs to be better cited

Also you simply the functions of dopamine, neuropeptide Y, Leptin and gherkin without tying them back to obesity and ED. You have a sentence but no literature that ties it back. 

Gut microbiota

You make several claims about gut microbiota, mood and behavior however provide no citations for it

Management

This section is significantly better than the last however, earlier you talk about ACEs and other sociological issues but don't address some of those in the management. What role can a physician play in these things?

What about the roles of people most likely to read this journal? (i.e. nutrition experts).

Author Response

We would like to thank the reviewer for the constructive comments on our manuscript. Changes made are all marked in yellow in the text.

  1. I understand that the prevalence of obesity and ED is high, but it would be great if you can provide statistics and cite them (i.e. ~165 million kids obese in the world). I know you mention it later in the manuscript, but it would be great to have it here as well)

Following the reviewer´s recommendation, we have included new statistics and cited them:

“…high prevalence, more than one hundred million [1].”

Prevalence of Obesity. World Obesity Federation:

 https://www.worldobesity.org/about/about-obesity/prevalence-of-obesity (Last visited 15/11/2021).

  1. In the introduction you talk about kids and adolescents. You should outline what ages you are covering in this review. I know you say pediatric but maybe a parenthetical reference would be great.

The definition of pediatric age is well established. In the present review the manuscripts included different periods of age from birth until 18 years. Each manuscript covers different age ranges however, they still remain between birth and 18 years.

  1. Cite lines 38-39

A new citation has been included, Hay P (4).

4.Hay P, Mitchison D. Eating Disorders and Obesity: The Challenge for Our Times. Nutrients. 2019;11:1055.

Obesity

  1. Cite lines 68-70

New citations have been included, Lee (8) and Di Cesare (9).

8.Lee EY, Yoon KH. Epidemic obesity in children and adolescents: risk factors and prevention. Front Med 2018,12:658-666.

  1. Di Cesare M, Sorić M, Bovet P, Miranda JJ, Bhutta Z, Stevens GA, Laxmaiah A, Kengne AP, Bentham J. The epidemiological burden of obesity in childhood: a worldwide epidemic requiring urgent action. BMC Med 2019,17:212.
  2. Cite lines 80-86

New citations have been included, Bucher [11] and [13] Falkner and Lurbe

11.Bucher Della Torre S, Keller A, Laure Depeyre J, Kruseman M. Sugar-Sweetened Beverages and Obesity Risk in Children and Adolescents: A Systematic Analysis on How Methodological Quality May Influence Conclusions. J Acad Nutr Diet 2016,116:638-59.

13.Falkner B, Lurbe E. Primordial Prevention of High Blood Pressure in Childhood: An Opportunity Not to be Missed. Hypertension 2020,75:1142-1150.

  1. In sociology and psychology literature socioeconomic adversity and family dysfunction have been linked fairly consistently to obesity. I think considering them as less conventional is minimizing the contributions of the two fields.

We agree with the reviewer, the sentence has been rewritten and the words “less conventional ones” have been removed:

“Along with the above well recognized factors, the presence of socioeconomic adversity, family dysfunction, offspring distress and junk food should be considered”

  1. You should also examine the association between ACEs and obesity as you refer to some of them later in the ED section. 

The following paragraph has been added including new bibliography (20,21)

“Adverse childhood experience (ACE) and its link with obesity has received more attention in the last years. In a recent study, children who had high intrafamilial adversity scores were more prone to be obese than children with low scores [20]. These results are in agreement with the findings of a meta-analysis of 41 studies in which the association of child maltreatment and obesity was assessed [21]. Nowadays, ACE is known to be a potentially modifiable risk factor for obesity.”

20.Kaufman J, Montalvo-Ortiz JL, Holbrook H, et al. Adverse Childhood Experiences, Epigenetic Measures, and Obesity in Youth. J Pediatr. 2018;202:150-156.e3.

21.Danese A, Tan M. Childhood maltreatment and obesity: systematic review and meta-analysis. Mol Psychiatry 2014;19:544–54.

  1. There need to be more citations in this section. You make several claims throughout this section that are not cited.

New citations have been added as per reviewers’ suggestion. Reference number: 4,8,9,11,13,19,20,21.

Eating Disorders

Overall this section was better cited than the obesity section. I would suggest investigating work that has examined ACEs a little more closely in this section.

A new paragraph including new bibliography has been added:

“Whether ACEs are true risk factors for the development of eating disorders remains unclear due to the scarcity of not only prospective studies but also the potential selection bias in clinical samples. In currently available studies, inconsistent results have been reported.  In a population based study, the authors stated that experiences of life events are associated with specific eating behaviors in children aged 10 years [36]. These findings sustain the fact that a link between adverse life events and emotional overeating exists [37, 38].”

36.Thomas R, Siliquini R, Hillegers MH, Jansen PW. The association of adverse life events with children's emotional overeating and restrained eating in a population-based cohort. Int J Eat Disord 2020,53:1709-1718.

37.Bjørklund O, Wichstrøm L, Llewellyn CH, Steinsbekk S. Emotional Over- and Undereating in Children: A Longitudinal Analysis of Child and Contextual Predictors. Child Dev 2019,90:e803-e818.

38.Michels N, Sioen I, Braet C, Eiben G, Hebestreit A, Huybrechts I, Vanaelst B, Vyncke K, De Henauw S. Stress, emotional eating behaviour and dietary patterns in children. Appetite 2012,59:762-9.

Environmental risk factors

This is a poorly cited section with multiple claims but no cited evidence to back it up.

This section has been reviewed and new citations have been added (43,44,45,46).

43.Libbey HP, Story MT, Neumark-Sztainer DR, Boutelle KN. Teasing, disordered eating behaviors, and psychological morbidities among overweight adolescents. Obesity (Silver Spring) 2008,16 Suppl 2:S24-9.

44.Macpherson-Sánchez AE. Integrating fundamental concepts of obesity and eating disorders: implications for the obesity epidemic. Am J Public Health 2015,105:e71-e85.

45.Field AE, Camargo CA Jr, Taylor CB, Berkey CS, Roberts SB, Colditz GA. Peer, parent, and media influences on the development of weight concerns and frequent dieting among preadolescent and adolescent girls and boys. Pediatrics 2001,107:54-60.

46.Striegel-Moore RH, Fairburn CG, Wilfley DE, Pike KM, Dohm FA, Kraemer HC. Toward an understanding of risk factors for binge-eating disorder in black and white women: a community-based case-control study. Psychol Med 2005,35:907–17.

Biological-genetic risk factors. Please provide examples of the genes that overlap.

According to the suggestions of the reviewer, 3 additional references have been included (48-51).  A paragraph concerning genes that overlap has been added:

“Genetic factors notably influence the regulation of neural circuits by controlling the appetite and satiety pathways, as well as the regulation of brain reward systems. Single Nucleotide Polymorphisms in genes linked to hypothalamic appetite and satiety mechanisms may be involved in the development of ED´s related to obesity such as BED and BN [51]”.

48.Castellini G, Franzago M, Bagnoli S, Lelli L, Balsamo M, Mancini M, Nacmias B, Ricca V, Sorbi S, Antonucci I, Stuppia L, Stanghellini G. Fat mass and obesity-associated gene (FTO) is associated to eating disorders susceptibility and moderates the expression of psychopathological traits. PLoS One 2017,12:e0173560.

49.Loos RJ, Bouchard C. FTO: the first gene contributing to common forms of human obesity. Obesity Reviews 2008,9:246–50.

50.Cameron JD, Tasca GA, Little J, Chyurlia L, Ritchie K, Yeh E, Doucette S, Obregon AM, Bulman DE, Doucet É, Goldfield GS. Effects of fat mass and obesity-associated (FTO) gene polymorphisms on binge eating in women with binge-eating disorder: The moderating influence of attachment style. Nutrition 2019, 61:208-212.

51.Nicoletti CF, Delfino HBP, Ferreira FC, Pinhel MAS, Nonino CB. Role of eating disorders-related polymorphisms in obesity pathophysiology. Rev Endocr Metab Disord 2019,20:115-125.

Psychological and personality risk factors. What are the different personality traits that may play a role? You state a large body of research suggests a link, please provide evidence.

 The revised sentence reads as follows:

[52]…Research suggests a link between emotional regulation and BE, as well as food addiction [53,54]. When negative affect and emotional dysregulation precede the occurrence of BE episodes, it exacerbates guilt and shame, creating a vicious cycle of losing control over eating [55].

New citations have been added (52,53,54).

52.Goldschmidt AB, Aspen VP, Sinton MM, Tanofsky-Kraff M, Wilfley DE. Disordered eating attitudes and behaviors in overweight youth. Obesity (Silver Spring) 2008,16:257-64.

53.Dingemans A, Danner U, Parks M. Emotion Regulation in Binge Eating Disorder: A Review. Nutrients 2017,9:1274.

54.Gearhardt AN, White MA, Masheb RM, Morgan PT, Crosby RD, Grilo CM. An examination of the food addiction construct in obese patients with binge eating disorder. Int J Eat Disord 2012,45:657-63.

Neuropsychogical and brain activity risk factors. Entire second paragraph is not cited

We have added the bibliography 58 accordingly. At the end of the section we have included the following sentence:

… [58]. A better understanding of wanting and linking mechanisms tailored to individual types of ED and obesity could lead to better therapeutic strategies, and perhaps help people who wish to more effectively create stop  signals to their own [57].

58.Berridge KC, Ho CY, Richard JM, DiFeliceantonio AG. The tempted brain eats: pleasure and desire circuits in obesity and eating disorders. Brain Res 2010,1350:43-64.

Behavioral and risk factors. Same as above. Poorly cited. Provide examples of where this happens (studies)

In agreement with the reviewer, new citations have been added (59,60). The following sentence has been included:

This has gained special attention during the COVID pandemic [59, 60].

59.Cecchetto C, Aiello M, Gentili C, Ionta S, Osimo SA. Increased emotional eating during COVID-19 associated with lockdown, psychological and social distress. Appetite 2021,160:105122.

60.Rodgers RF, Lombardo C, Cerolini S, Franko DL, Omori M, Fuller-Tyszkiewicz M, Linardon J, Courtet P, Guillaume S. The impact of the COVID-19 pandemic on eating disorder risk and symptoms. Int J Eat Disord 2020,53:1166-1170.

Biochemical. Same as above section. It needs to be better cited. Also you simply the functions of dopamine, neuropeptide Y, Leptin and gherkin without tying them back to obesity and ED. You have a sentence but no literature that ties it back. 

New citations have been added. 61,62,64,66,68

61.Gross MJ, Kahn JP, Laxenaire M, Nicolas JP, Burlet C. Corticotropin-releasing factor and anorexia nervosa: reactions of the hypothalamus-pituitary-adrenal axis to neurotropic stress. Ann Endocrinol (Paris) 1994,55:221-8.

62.Licinio J, Wong ML, Gold PW. The hypothalamic-pituitary-adrenal axis in anorexia nervosa. Psychiatry Res 1996,62:75-83.

64.Pruccoli J, Parmeggiani A, Cordelli DM, Lanari M. The role of Noradrenergic system in eating disorders: A systematic review. Int J Mol Sci 2021,22:11086.

66.Carlson N. "Ingestive Behavior". Physiology of Behavior. University of Massachusetts, Amherst: Pearson. pp. 428–432. ISBN 978-0-205-23939-9. 2013

68.Atalayer D, Gibson C, Konopacka A, Geliebter A. Ghrelin and eating disorders. Prog Neuropsychopharmacol Biol Psychiatry 2013,40:70-82.

Gut microbiota. You make several claims about gut microbiota, mood and behavior however provide no citations for it

A new citation has been added (70).

70.Biedermann, L., Rogler, G. The intestinal microbiota: its role in health and disease. Eur J Pediatr 2015,174:151–167.

Management

This section is significantly better than the last however, earlier you talk about ACEs and other sociological issues but don't address some of those in the management. What role can a physician play in these things?

A new paragraph and bibliography has been included. (92,93)

5.7 Screening for Adverse Childhood Experiences

There is limited data on the effect of structured intervention models to treat obesity and EDs in children that have experienced ACEs. Since there is an association of ACEs with obesity and EDs, the healthcare team should consider the possibility of having an ACE. Screening for ACEs would be regularly performed using validated tools [92, 93]. In the case of an affected child, multicomponent intervention startegies should include appropriate psycho-social support and counseling to manage anxiety due to trauma that would empede the effectiveness of obesity and ED treatment. Of note, prevention policies for ACEs that may echance self confidence, social and emotional skills could accompany healthy eating education on an indivdual and family level.

92.Chu WW, Chu NF. Adverse childhood experiences and development of obesity and diabetes in adulthood-A mini review. Obes Res Clin Pract 2021,15:101-105.

93.Isohookana R, Marttunen M, Hakko H, Riipinen P, Riala K. The impact of adverse childhood experiences on obesity and unhealthy weight control behaviors among adolescents. Compr Psychiatry 2016,71:17-24.

What about the roles of people most likely to read this journal? (i.e. nutrition experts).

A multidisciplinary approach is most beneficial for the child and family as it allows for multiple points of view and sources of knowledge to come together in attempting to help this vulnerable population (children and adolescents) with their health in the short and long term. The well-being of the child encompasses not only physical health but also emotional and mental health. Due to this, the entire healthcare team should be involved.

Reviewer 2 Report

Stabouli et al., provided a well-organized review article about obesity and eating disorders in children and adolescents. Although there is no big problem in the manuscript, they should modify following points.

1: the authors explain MC4R in lines 103-104 together with FTO in 4.2 Risk factors (lines 161-165).  

2: In lines 130-, “Emerging evidence…”, the authors should explain what kind of evidence (or example) show relationship between food insecurity and bulimic-spectrum among adults.

3: Lines 184-186, please cite corresponding papers.
4: Lines 195-197, please cite corresponding papers.
5: Line 302, the authors should briefly explain what Orlistat is.

Author Response

We would like to thank the reviewer for the constructive comments on our manuscript. Changes made are all marked in yellow in the text.

Stabouli et al., provided a well-organized review article about obesity and eating disorders in children and adolescents. Although there is no big problem in the manuscript, they should modify following points.

1: The authors explain MC4R in lines 103-104 together with FTO in 4.2 Risk factors (lines 161-165).  

As suggested by the reviewer, the connection of these two paragraphs has been addressed. A sentence has been added after lines 161-165. Bibliography 19 has been added.

“In addition, as already mentioned before, a mutation in MC4R is the most common gene defect associated with an early form of obesity in children (19)”

19.Loos RJF, Yeo GSH. The genetics of obesity: from discovery to biology. Nat Rev Genet 2021,23:1–14. doi: 10.1038/s41576-021-00414-z. Epub ahead of print.

2: In lines 130-, “Emerging evidence…”, the authors should explain what kind of evidence (or example) show relationship between food insecurity and bulimic-spectrum among adults.

An example has been added and the corresponding paragraph has been rewritten. Furthermore, a new bibliography has been added.

“A new element has come into play which is food insecurity, characterized by limited or uncertain means of accessing nutritious food in a safe and socially acceptable manner.  Emerging evidence consistently indicates that food insecurity is cross-sectionally associated with the bulimic-spectrum among adults. This has been shown in a national representative sample of US adults. During a 12 months period diagnoses of bulimic –spectrum  disorders, mood  disorders, and anxiety disorders were more common among individuals who have experienced food insecurity than among who were food secure.  The study highlighted that the greatest difference was observed for bulimic-spectrum eating disorders [35]. Considering these findings, it may be necessary to take the pediatric population into consideration. This emerging evidence needs much more research to better understand this issue”

35.Hazzard VM, Barry MR, Leung CW, Sonneville KR, Wonderlich SA, Crosby RD. Food insecurity and its associations with bulimic-spectrum eating disorders, mood disorders, and anxiety disorders in a nationally representative sample of U.S. adults. Soc Psychiatry Psychiatr Epidemiol 2021,27:1–8. doi: 10.1007/s00127-021-02126-5. Epub ahead of print.

3: Lines 184-186, please cite corresponding papers.

The query has also been addressed by reviewer 1. We have added the bibliography accordingly, 58. At the end of the section we have included the following sentence:

“… [58]. A better understanding of wanting and linking mechanisms tailored to individual types of ED and obesity could lead to better therapeutic strategies, and perhaps help people who wish to more effectively create stop  signals to their own [57].”

58.Berridge KC, Ho CY, Richard JM, DiFeliceantonio AG. The tempted brain eats: pleasure and desire circuits in obesity and eating disorders. Brain Res 2010,1350:43-64.4: Lines 195-197, please cite corresponding papers.

In agreement with the reviewer, new citations have been added (59,60). The following sentence has been included:

“This has gained special attention duirng the COVID pandemic [59, 60].”

59.Cecchetto C, Aiello M, Gentili C, Ionta S, Osimo SA. Increased emotional eating during COVID-19 associated with lockdown, psychological and social distress. Appetite 2021,160:105122.

60.Rodgers RF, Lombardo C, Cerolini S, Franko DL, Omori M, Fuller-Tyszkiewicz M, Linardon J, Courtet P, Guillaume S. The impact of the COVID-19 pandemic on eating disorder risk and symptoms. Int J Eat Disord 2020,53:1166-1170.5: Line 302, the authors should briefly explain what Orlistat is.

The following sentence has been included:

“Orlistat is the only available anti-obesity drug that does not involve the mechanisms of appetite. It induces weight reduction via the inhibition of lipases in the mucous membranes of the stomach, small intestine, and pancreas, thereby preventing the breakdown of triglycerides into fatty acids and their absorption in the intestines [88]”

88.Son JW, Kim S. Comprehensive Review of Current and Upcoming Anti-Obesity Drugs. Diabetes Metab J 2020,44:802-818.

Round 2

Reviewer 1 Report

I'd like to thank the authors for taking the time to make the corrections. Just a few more corrections.

You provide a plethora of biochemical factors that may influence EDs however, you still only provide simple explanations of what those hormones/neurotransmitters do and not how they could be related to ED or obesity (except for leptin and ghrelin). Please expand upon the relationship between noradrenergic system and ED.

Author Response

We thank the reviewer for the comments. We have added the following information, page 8, last paragraph:

Besides its relevant direct, hypotalamus-based actions on feeding regulation the noradrenergic system is indirectly implied in various endocrine networks controlling human nutrition.